# Noise resilient exceptional-point voltmeters enabled by oscillation quenching phenomena

Arunn Suntharalingam [1], Lucas Fernández-Alcázar [2,3], Rodion Kononchuk [1] & Tsampikos Kottos [1] ✉

Exceptional point degeneracies (EPD) of linear non-Hermitian systems have been recently utilized for hypersensitive sensing. This proposal exploits the sublinear response that the degenerate frequencies experience once the system is externally perturbed. The enhanced sensitivity, however, might be offset by excess (fundamental and/or technical) noise. Here, we developed a self-oscillating nonlinear platform that supports transitions between two distinct oscillation quenching mechanisms – one having a spatially symmetric steady-state, and the other with an asymmetric steady-state – and displays nonlinear EPDs (NLEPDs) that can be employed for noise-resilient sensing. The experimental setup incorporates a nonlinear electronic dimer with voltage-sensitive coupling and demonstrates two-orders signal-to-noise enhancement of voltage variation measurements near NLEPDs. Our results resolve a long-standing debate on the efficacy of EPD-sensing in active systems above self-oscillating threshold.

The underlying mathematical structures of non-Hermitian wave systems[1–4] have inspired the last few years new technologies[5–8]. Many of these are reliant on the existence of exceptional point degeneracies (EPDs)[8]. These are non-Hermitian degeneracies where a set of $N$ eigenvalues and their corresponding eigenvectors coalesce[1,2]. In the proximity of an $N$th order EPD (EPD-N), the eigenvalue detuning $\Delta f \equiv |f - f_{EPD}|$, due to a small external perturbation $\varepsilon$, follows a sublinear response (SLR) $\Delta f \sim \sqrt[N]{\varepsilon} \gg \varepsilon$ that can be utilized for enhanced sensing[9–15].

A principal requirement for efficient EPD sensing is the increase of the resolution limit via the narrowing of the resonance linewidth. This can be achieved by judicious design of cavity amplification mechanisms. The downside of this strategy is that it introduces additional noise that, in some EPD platforms, might offset the enhanced signal sensitivity[13,16–19]. Furthermore, nonlinear effects might become important—requesting the development of theoretical tools that treat them on equal footing with the sensitivity enhancement near EPDs. However, most current studies rely on linear mathematical constructs, such as the Petermann factor[20–22], which describes the linewidth enhancement near EPDs due to the biorthogonal nature of the eigenmodes of the underlying linear non-Hermitian systems[16]. Obviously, this approach is not suitable when the response of a system is influenced by nonlinearities. An example case is a laser at an EPD. Fortunately, an appropriate language exists from the area of dynamical systems and bifurcation theory[23–27], which can be adopted for the analysis of nonlinear EPDs (NLEPDs). Examples of systems that are amenable to such analysis are shown in Fig. 1a. In fact, some recent theoretical studies have utilized this approach to address issues like the formation of NLEPDs and the emulation of neuronal dynamic functions using parity-time ($\mathcal{PT}$) symmetric systems that involve gain and loss nonlinear channels[26,27]. These neuromorphic functionalities, belong to the general category of oscillation quenching mechanisms, whose characteristics are determined by the underlying dynamical symmetries of the system[28–30]. Oscillation quenching mechanisms occur in systems of coupled nonlinear oscillators, and can be classified in two different

[1]Wave Transport in Complex Systems Lab, Department of Physics, Wesleyan University, Middletown, CT, USA. [2]Institute for Modeling and Innovative Technology, IMIT (CONICET - UNNE), W3404AAS Corrientes, Argentina. [3]Physics Department, Natural and Exact Science Faculty, Northeastern University of Argentina, W3404AAS Corrientes, Argentina. ✉e-mail: tkottos@wesleyan.edu

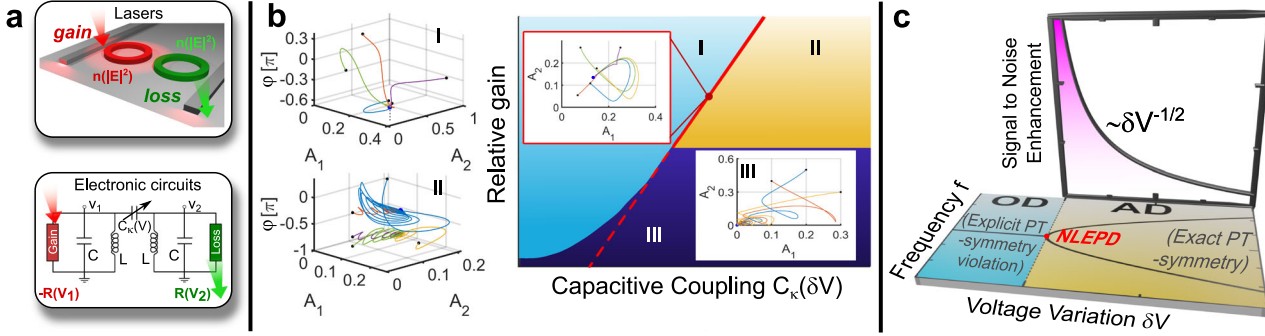

**Fig. 1 | EPDs at the transition between OD and AD. a** Physical systems that demonstrate oscillation death (OD) and amplitude death (AD) oscillation quenching. **b** (Right) The parameter space of the circuit in (**a**), is partitioned in three distinct domains that host stable nonlinear supermodes (NS) (fixed points of the dynamical equations associated with Jacobian eigenvalues $\{\lambda_n; n = 1,2,3\}$ with $Re(\lambda_n) < 0$) with distinct dynamical symmetries. Transition from one domain to another is dictated by the relative gain $\gamma_1^{(0)}/\gamma_2^{(0)}$ and voltage variation $\delta V$ that controls the capacitive coupling between the two nonlinear $RLC$ tanks. The three domains are: (I) the OD domain, (II) the AD domain, and (III) the trivial steady-state solution domain. The steady-state field amplitude $A_n$ of each resonator $n = 1,2$ differs from one another in domain (I) while it is the same in domain (II). Typical examples of $(A_1, A_2, \varphi)$-phase-space trajectories, $\varphi$ being the relative phase, are shown on the subfigures. The black dots indicate initial conditions while the blue dots the steady-state (fixed point). The red solid line indicates nonlinear exceptional point degeneracies (NLEPDs) associated with the coalescence of two stable NS. The red dashed line indicates NLEPDs associated with the coalescence of two non-trivial unstable NS (see Supplementary Note 5 and Supplementary Fig. 3). **c** (Horizontal plane) Parametric evolution of the nonlinear eigenfrequencies $f$ versus $\delta V$ for a fixed value of the relative gain corresponding to a transition from OD to AD via a NLEPD, see red dot. (Vertical plane) The signal-to-noise enhancement factor, SNE $= \frac{\chi}{\alpha_{VRW}}$ (where $\chi \equiv \frac{\partial\left(\frac{\Delta f_+}{f_0}\right)}{\partial(\delta V)}$ is the sensitivity and $\alpha_{VRW}$ is the noise-equivalent voltage variations), diverges as $\frac{1}{\sqrt{\delta V}}$ in the proximity of NLEPD.

types[30]: oscillation death (OD), which is associated with the formation of an inhomogeneous steady state, and amplitude death (AD), which results in a homogenous steady state. These mechanisms have been observed in a variety of dynamical systems ranging from climate, lasers, electronic circuits, chemical systems, neurons, and more (for indicative references, see review paper[30]). Recently, in the framework of non-Hermitian photonic systems with $\mathscr{PT}$-symmetry, oscillation quenching mechanisms have been proposed, theoretically, for topological protection, signal processing[26,27] and memory devices[31]. The interplay of $\mathscr{PT}$-symmetry and oscillation quenching were also theoretically discussed in the framework of electronic circuits. These theoretical studies highlighted a connection between a broken (exact) $\mathscr{PT}$-symmetric phase and an OD (AD) oscillation quenching mechanism, see Fig. 1b.I and Fig. 1b.II, respectively. The transition between these two regimes is characterized by the formation of a NLEPD (see Fig. 1b right and Fig. 1c). Can these NLEPDs be used for sensing and what is the signal-to-noise enhancement (SNE) factor in their proximity? A definite answer to this question requires not only a theoretical modeling[24,25,32–36], but, most importantly, the establishment of controllable experimental platforms that will scrutinize the predictions of the theory, and guide the theoretical language as it is developing.

Here, we address the viability of NLEPD sensing protocols using two nonlinear RLC tanks whose capacitive coupling is used as a sensing platform for voltage variations. The RLC circuits have anharmonic parts consisting of a complementary amplifier (gain) and a dissipative conductor (loss), see Fig. 1a and "Methods." The nonlinear supermodes (NS) are the fixed points of the dynamical equations, and their properties arise from the underlying dynamical symmetries and stability. We focus on stable NS that are experimentally accessible and these are identified from the (negative) real part of the eigenvalues $\{\lambda_n\}$ of the Jacobian matrix—which describes the linearized dynamics around each of these fixed points[23]. Their properties lead to the partition of the parameter space into three distinct domains (see domains I, II, III in Fig. 1b right). The last of them, domain III, involves trivial NS with zero amplitude at each RLC tank (see inset of right subfigure of Fig. 1b) and is therefore irrelevant to our investigations. The other two domains, are separated by a NLEPD (see red line in right subfigure of Fig. 1b and

red point in Fig. 1c) and contain one (two) non-trivial stable hyperbolic fixed points in the OD (AD) phase, see Fig. 1c (and Fig. 1b.I and Fig. 1b.II, respectively). The stable NS in the AD phase coalesce at the NLEPD-point at a voltage variation $\delta V = 0$, where $\delta V$ controls the capacitive coupling between the two resonators. The detuned eigenfrequencies follow a characteristic SLR, $\Delta f_\pm \equiv f_\pm - f_{NLEPD} \propto \pm\sqrt{\delta V}$, leading to two-orders enhancement of sensitivity to small voltage variations, and a similar SNE near the NLEPD, see Fig. 1c. Our results challenge the validity of linear concepts (e.g., Petermann factor) for the noise analysis near NLEPDs, and confirm beyond doubt that self-oscillating systems above threshold have an enhanced signal-to-noise sensing performance in the proximity of the NLEPDs.

## Results
### Experimental platform
The sensor consists of a pair of nonlinear RLC resonators[37–40] (see Fig. 1a) with natural frequency $f_0 = \frac{1}{2\pi}\frac{1}{\sqrt{LC}} \approx 338$ kHz, and an impedance (at resonance) $Z_0 = \sqrt{\frac{L}{C}} \approx 424\,\Omega$. One of the resonators (gain−indicated with red in Fig. 1a) incorporates a nonlinear amplifier, $−R_1(V_1)$, which is characterized by an $I$−$V$ curve of $I_1(V_1) = −\frac{V_1}{R_1^{(0)}} + bV_1^3$ while the other one (loss−indicated with green in Fig. 1a) incorporates a nonlinear loss, $R_2(V_2)$, with an $I$−$V$ curve of $I_2(V_2) = \frac{V_2}{R_2^{(0)}} + bV_2^3$ (where $b \approx 7 \cdot 10^{-4}$ AV$^{-3}$ and $V_{1(2)}$ are the voltages at the nodes 1(2)). The two resonators are coupled together via a capacitance voltage controlled (CVC) capacitor $C_\kappa(V) = \kappa \cdot C$ where $\kappa$ is a dimensionless parameter representing the strength of the coupling. The linear conductances, $\frac{1}{R_1^{(0)}} > \frac{1}{R_2^{(0)}}$, were tuned such that the system undergoes a transition from AD to OD as the voltage at the coupling capacitor varies (see Fig. 1b, c and "Methods"). A transmission line (TL) with impedance $z_0 = 50\,\Omega$ is weakly coupled to each resonator via capacitors $C_e \ll C$. The TLs were used to collect and direct the signal generated by the circuit to a VNA for further processing (see "Methods"). The NLEPD (occurring at $\delta V = 0$) can be experimentally identified as the point for which the voltage amplitudes $\frac{V_1}{V_2}$ of each RLC resonator deviates from unity (AD domain) and begins to acquire larger values (OD domain)—see Fig. 2b.

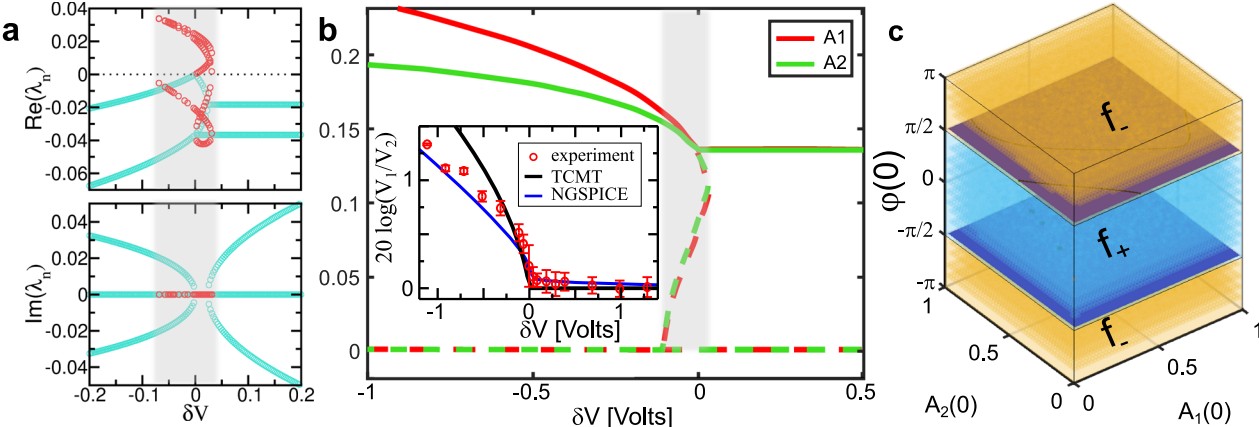

**Fig. 2 | Stability analysis and basins of attraction. a** Eigenvalues $\lambda_n$ of the Jacobian evaluated at the stable (unstable) nontrivial nonlinear supermodes (NS), indicated with turquoise (red) circles. **b** Field amplitude $A_1(A_2)$ of the NS versus voltage variations $\delta V$. The solid lines indicate stable fixed points (see Eq. (2)) while the dashed lines indicate unstable solutions evaluated numerically using Eq. (1) together with the eigenvalue analysis of the Jacobian matrix in panel **a** (see also Supplementary Note 5). The fixed relative gain $\frac{\gamma_1^{(0)}}{\gamma_2^{(0)}} = 1.46$ is chosen in a way that the system undergoes a transition from oscillation death (OD) to amplitude death (AD) as the voltage variation $\delta V$ increases. The nonlinear exceptional point degeneracy (NLEPD) occurs at $\delta V = 0$. (Inset) Measured average logarithmic voltage ratio (red

symbols) $\frac{V_1}{V_2}$ of the NS. Each point represents an average of five independent measurements and the error bars indicate ±1 standard deviation. The black line is the prediction of temporal coupled mode theory (TCMT). The **b**lue line indicates the results from NGSPICE. In panels **a** and **b**, the gray shadow highlights the presence of nontrivial unstable NS. **c** Phase-space analysis and basins of attraction for the stable fixed points associated with the upper ($f_+$) (blue highlighted domain) and lower ($f_-$) (yellow highlighted domain) branches of the NS in the AD domain. The voltage variation is $\delta V \approx 2$ mV corresponding to a circuit configuration in the proximity of the NLEPD.

## Theoretical analysis of nonlinear supermodes

The voltage dynamics $V_n$ at each RLC tank ($n = 1,2$), is described via a temporal coupled mode theory (TCMT) (see Supplementary Notes 1 and 2)

$$i\frac{d}{d\tau}\begin{pmatrix} a_1 \\ a_2 \end{pmatrix} = \begin{pmatrix} \nu_\kappa + i\gamma_1 & \frac{\kappa}{2} \\ \frac{\kappa}{2} & \nu_\kappa - i\gamma_2 \end{pmatrix}\begin{pmatrix} a_1 \\ a_2 \end{pmatrix}; \quad (1)$$

where $a_n \equiv \sqrt{\frac{3}{8}bZ_0}\left(V_n + i\frac{1}{2\pi}\frac{\dot{V}_n}{f_0}\right), \gamma_n = \gamma_n^{(0)} + (-1)^n(|a_n|^2 + \eta), \nu_\kappa = 1 - \frac{\kappa}{2} - \sqrt{\frac{\eta}{2}\frac{Z_0}{z_0}}$ and $\tau \equiv 2\pi f_0 t$ is the rescaled time. The parameters $\gamma_n^{(0)}$ are the linear gain ($n = 1$) and loss ($n = 2$) coefficients associated with the gain and loss RLC resonator, respectively, while $\eta = \frac{1}{2}\frac{z_0}{Z_0}\left(\frac{c_c}{c}\right)^2$ models the coupling of the circuit to the TLs. The global frequency shift $f_\kappa = f_0 \cdot \nu_\kappa$ is associated with the renormalization of the natural frequency $f_0$ of the RLC resonators due to the capacitive coupling between them ($\kappa$-term) and the coupling with the TLs ($\eta$-term). The $\kappa$-dependence could be, in principle, avoided if we choose another type of coupling (e.g., inductive coupling). Below, we analyze the steady-state properties of Eq. (1) in terms of the coupling parameter $\kappa = \kappa(\delta V)$, which is used as a sensing platform of voltage variations $\delta V$ (see "Methods").

The nonlinearities in our system have been chosen carefully to prevent the system from evolving towards undesirable unbounded states where $A_1$ and/or $A_2 \to \infty$. This can be easily realized from Eq. (1) by recognizing that whenever the field intensity of the gain resonator exceeds a critical value $|a_1|^2 > \gamma_1^{(0)} - \eta$ the gain coefficient $\gamma_1$ becomes negative, thus turning the gain RLC tank into a lossy one. The NS of Eq. (1) may be expressed in the polar representation $a_n = A_n e^{i\varphi_n}e^{-\frac{if\tau}{f_0}}$ and are identified as the fixed points of the dynamical system whose evolution is defined in a three-dimensional phase space ($A_1, A_2, \varphi \equiv \varphi_2 - \varphi_1$). We classify these fixed points according to their underlying (dynamical) symmetry, and stability. The latter is determined by the eigenvalues $\{\lambda_1, \lambda_2, \lambda_3\}$ of the $3 \times 3$ Jacobian matrix, $J$, when evaluated at the fixed point (see Supplementary Note 5)[23]. When $Re(\lambda_n) \neq 0$ ($\forall n = 1,2,3$), the fixed point is a hyperbolic equilibrium and there is a homeomorphism that maps the phase portrait in its proximity onto

solutions of its linearized system described by $J$[41]. When all $Re(\lambda_n) < 0$, the fixed point is stable, and it is unstable if at least one $Re(\lambda_n) > 0$. Hyperbolic equilibria are robust to small variations which do not, qualitatively, change the phase portrait. The opposite scenario of non-hyperbolic equilibria is associated with cases where either one of the eigenvalues of the Jacobian matrix is zero or has a zero real part. These are structurally unstable cases, and one can numerically test the nature of the stability of these fixed points by direct dynamical simulations with Eq. (1).

We have found that one fixed point of Eq. (1) is a trivial state ($A_1, A_2$) = (0,0). The dynamical simulations indicate that it is stable in the parametric domain III (see Fig. 1b) while it is unstable in the other two domains. Below, we analyze the stable non-trivial fixed points occurring in domains I and II of Fig. 1b. In these cases, the real-valued amplitudes $A_n > 0$ take the form (see Supplementary Note 4 and Supplementary Fig. 2):

$$A_n = \rho^{n-1} \cdot \sqrt{\gamma_1^{(0)} - \eta - \frac{\kappa\rho}{2}} \text{ for } \kappa \leq \gamma_1^{(0)} + \gamma_2^{(0)}(\text{domain I});$$
$$A_1^{(\pm)} = A_2^{(\pm)} = \sqrt{\frac{\gamma_1^{(0)} - \gamma_2^{(0)} - 2\eta}{2}} \text{ for } \kappa \geq \gamma_1^{(0)} + \gamma_2^{(0)}(\text{domain II}); \quad (2)$$

where the real-valued variable $\rho \equiv \frac{A_2}{A_1} > 0$ is a solution of the quartic algebraic equation $1 - 2\rho\left(\frac{\gamma_2^{(0)} + \eta}{\kappa}\right) - 2\rho^3\left(\frac{\gamma_1^{(0)} - \eta}{\kappa}\right) + \rho^4 = 0$. The physical requirement $A_{1,2}^{(\pm)} \in \Re(A_n \in \Re)$ leads to the condition $\gamma_1^{(0)} - \gamma_2^{(0)} - 2\eta \geq 0$ ($\gamma_1^{(0)} - \eta - \frac{\kappa\rho}{2} \geq 0$) which determines the boundary between domains II and III (domains I and III), see Fig. 1b. Finally, the relation $\kappa_{NLEPD} = \gamma_1^{(0)} + \gamma_2^{(0)}$ defines the transition between domains I and II− which is characterized by the formation of a NLEPD associated with the coalescence of two stable NS (see solid red line in Fig. 1b). This is further confirmed by evaluating the nonlinear eigenfrequencies $f_\pm(\kappa)$ associated with the solutions of Eq. (2). From the TCMT, we have that (see Supplementary Note 4a, b):

$$f_\pm = \begin{cases} f_\kappa & \text{for } \kappa \leq \kappa_{NLEPD}(\text{domain I}); \\ f_\kappa \pm \frac{f_0}{2}\sqrt{\kappa^2 - \kappa_{NLEPD}^2} & \text{for } \kappa \geq \kappa_{NLEPD}(\text{domain II}); \end{cases} \quad (3)$$

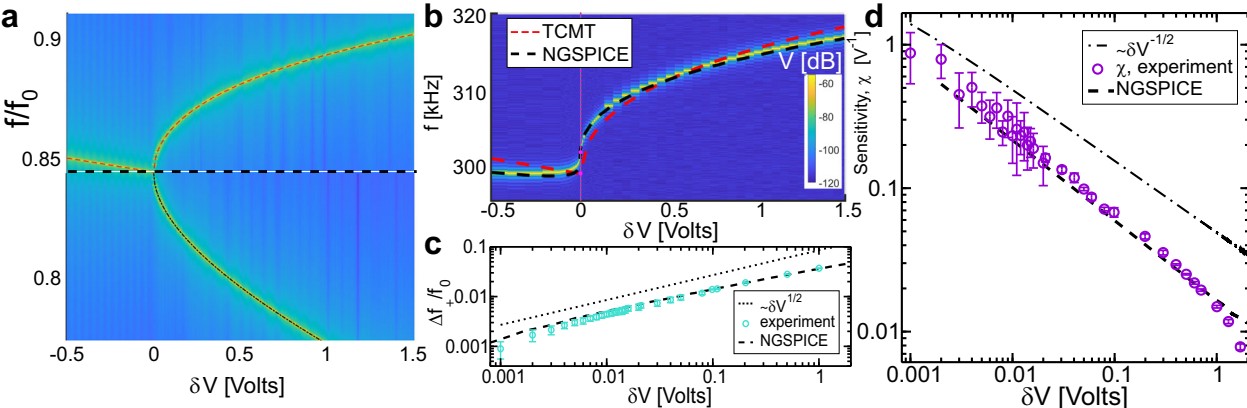

**Fig. 3 | Sublinear frequency detuning and sensitivity to applied voltage variations. a** Density plot of the normalized emitted spectrum evaluated from dynamical simulations of the temporal coupled mode theory (TCMT) model versus voltage variations of the coupling capacitor. The nonlinear frequencies for $\delta V > 0$ have been obtained using different initial conditions which belong to the basin of attraction of the corresponding stable fixed point. The dashed black line indicates the frequency domain for which each initial condition has been used.
**b** Experimentally measured emitted spectrum as a function of voltage variations. The red dashed line in both **a** and **b** is the TCMT prediction Eq. (3) of the nonlinear frequencies. The magenta dotted line indicates the position $\delta V = 0$, where the

nonlinear exceptional point degeneracy (NLEPD) is located. **c** The measured relative frequency detunings (circles) for the stable fixed point associated with the upper branch versus the applied voltage variations. The black dashed line is drawn to guide the eye and has a slope of $\frac{1}{2}$ that is characteristic of a NLEPD of order $N = 2$.
**d** The sensitivity of the active nonlinear circuit demonstrating two orders enhancement in the proximity of the NLEPD as opposed to a system configuration away from the NLEPD. The black dashed line in panels **b**, **c** and **d** indicates the numerical results using NGSPICE. Error bars in panels **c** and **d** indicate ±1 standard deviation obtained from ten independent measurements.

where the square-root dependence of the eigenfrequencies from the coupling detuning $\kappa$ reflects the presence of the NLEPD. In the range of voltage variations that have been used in our experiment ($-1.1\,\text{V} \leq \delta V \leq 2\,\text{V}$ with resolution of 1 mV), the coupling is $\kappa(\delta V) \approx \kappa_{\text{NLEPD}} - (0.0234\,\text{V}^{-1})\cdot\delta V$ with $\kappa_{\text{NLEPD}} \equiv \gamma_1^{(0)} + \gamma_2^{(0)} \approx 0.30$ (where $\gamma_1^{(0)} \approx 0.18$; $\gamma_2^{(0)} \approx 0.12$).

The two fixed points in domain II ($\kappa \geq \kappa_{\text{NLEPD}}$), are associated with the AD phase where the field amplitudes at each resonator are the same and the two stable NS differ only by the relative phase $\varphi_{\pm}$ (see Supplementary Note 4a). In this parameter range, the system respects an exact parity-time symmetry−where both the system and the corresponding NS are invariant under a joint parity (i.e., space inversion 1 ↔ 2) and time-reversal (i.e., complex conjugation) symmetry. In domain I ($\kappa \leq \kappa_{\text{NLEPD}}$), instead, there is only one non-trivial stable NS. This domain is associated with the OD phase where the field amplitudes at each RLC resonator differ from one another. A detailed fixed point numerical analysis using a MATLAB fsolve routine confirms the above theoretical predictions and provides more general information about other (unstable) fixed points, as well as the stability analysis of the nontrivial NS via the eigenvalues of the Jacobian (see Fig. 2a, b). In the inset of Fig. 2b, we also report some represented values of the measured voltage ratios $\frac{V_1}{V_2}$ versus the voltage variation $\delta V$. These results compare nicely with the TCMT predictions (black line) indicating that the NLEPD occurs at the transition between AD and OD phases. The slight deviations for large negative $\delta V$ are attributed to the small detunings of various components in our circuit from its ideal (TCMT) parameters (see Supplementary Note 7 and Supplementary Fig. 4) and to the limitations of the TCMT. The applicability of the latter is subject to a number of approximations: high-$Q$ factors of each RLC resonator, weak coupling between the two RLC tanks, and the elimination of fast-oscillating terms (see Supplementary Note 1). Nevertheless, the overall agreement between the predictions of TCMT, the NGSPICE simulations (blue solid line) and the experimental results are satisfactory. Additionally, the TCMT predicts that the voltage contrast follows a Puiseux expansion, $\frac{V_1}{V_2} \approx 1 + |\delta V|^{\frac{1}{3}} + O(|\delta V|^{\frac{4}{3}})$, resulting in $\log\left(\frac{V_1}{V_2}\right) \sim |\delta V|^{\frac{2}{3}}$. This sublinear response of the voltage contrast suggests that it could also be used as another physical observable for sensing purposes. Further analysis on the SNE of such observable is necessary to establish it as sensing measurand and will be reported elsewhere.

The existence of a stable NS does not guarantee the evolution of the system to this specific state. Instead, the system may evolve either to a stable trivial state, or to another stable NS in case of bistabilities (AD domain). The former scenario is easily excluded by an appropriate choice of the relative gain parameter (see Fig. 1b). The latter scenario can be controlled by realizing that the final state depends strongly on the initial conditions $\{A_1(0), A_2(0), \varphi(0)\}$ (see Supplementary Note 6). The phase-space volume that contains the initial conditions which converge to a specific stable fixed point constitute a basin of attraction, and its size provides a measure of how attractive this fixed point is. Detailed dynamical simulations using Eq. (1)−for various $\kappa$ values along with a fine mesh of initial conditions $\{A_1, A_2, \varphi\}$−allowed us to identify the basins of attraction of the two fixed points in the AD domain. We find that initial excitations with a relative phase $|\varphi| > \frac{\pi}{2}$ end up at the AD fixed point associated with the $f_-$ mode while an initial preparation with a phase $|\varphi| < \frac{\pi}{2}$ leads to a $f_+$ supermode. In Fig. 2c, we show the basins of attraction for the $f_+(f_-)$ fixed points which are indicated with blue (yellow) color for the example case of $\delta\kappa \equiv \kappa(\delta V) - \kappa_{\text{NLEPD}} \approx 5 \cdot 10^{-5}$ (corresponding to $\delta V \approx 2\,\text{mV}$). In fact, further analysis using TCMT indicated that a small detuning between the resonant frequencies of the two RLC resonators smoothens the frequency splitting in the close proximity of the NLEPD without drastically affecting the square-root response of $f_+$. At the same time, it has important consequences on the stability of the NS in the vicinity of the NLEPD by favoring only the upper branch $f_+$ which remains stable−as opposed to the lower branch $f_-$ that turns unstable (see Supplementary Note 7 and Supplementary Fig. 4). Away from the NLEPD the bistable nature at the AD domain persists. Either way, the square-root scaling of the NS frequency $f_+$ from $f_{\text{NLEPD}}$ (see Eq. (3)) is unaffected.

**Sensing protocol**

In Fig. 3a, we report a density-plot of the voltage power spectrum $|V_1(\omega)|^2$ for various voltage variations $\delta V$ by performing a Fourier transform of the temporal field−which were evaluated via time-domain simulations of the TCMT Eq. (1). To achieve the asymptotic states associated with the $f_+(f_-)$ supermodes in the AD domain, we have prepared the initial excitation at relative phase $|\varphi| < \frac{\pi}{2} (|\varphi| > \frac{\pi}{2})$ as discussed above. The numerical data agree nicely with the theoretical predictions of Eq. (3). In Fig. 3b, we show a density plot of the

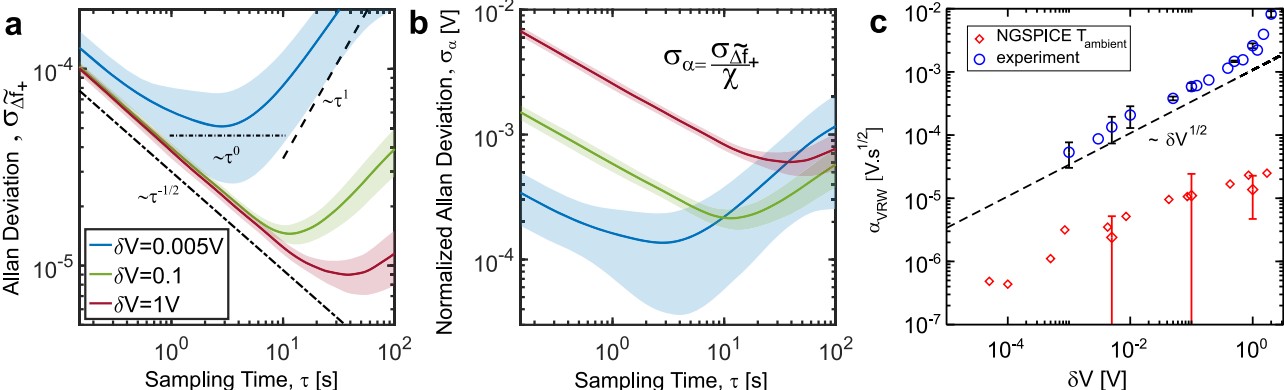

**Fig. 4 | Noise analysis at various voltage variations. a** The Allan deviation $\sigma_{\widetilde{\Delta f_+}}(\tau)$ of the circuit readout versus the sampling time $\tau$ is measured at various voltage variations $\delta V$ of the coupling capacitor both in the proximity (small $\delta V$ values) and away (large $\delta V$ values) from the nonlinear exceptional point degeneracy (NLEPD). **b** The rescaled (with respect to the sensitivity $\chi$) Allan deviation $\sigma_\alpha(\tau) = \sigma_{\widetilde{\Delta f_+}}(\tau)/\chi$ decreases as we are approaching the NLEPD indicating that the sensitivity enhancement offsets the noise enhancement. **c** The measured (blue circles) voltage random walk coefficient $\alpha_{VRW}$ versus the voltage variations for all $\delta V$ voltage variations that we have used. The red diamonds are the results of NGSPICE simulations where we have considered thermal (Johnson–Nyquist) noise at the resistors, amplifiers and at the transmission lines (TLs) described by an ambient temperature $T = 300$ K. In all cases, the error bars are indicated by the shadow area and correspond to $\pm 1$ standard deviation evaluated over four different measurements.

measured power spectrum of the emitted signal together with the TCMT predictions of Eq. (3) for the $f_+$ frequency (red dashed line). The absence of $f_-$ from the measured power spectrum, is associated with the fact that the experimental initial preparation favors a field excitation with a small relative phase $|\varphi| < \frac{\pi}{2}$. In the same figure, we also report the $f_+$ frequency (versus $\delta V$) that has been extracted by a Fourier transform of the voltage $V_1(t)$ using NGSPICE simulations (black dotted line).

The sublinear detuning is better appreciated by reporting $\Delta f_+ \equiv f_+ - f_{NLEPD}$ versus $\delta V$ in a double-logarithmic plot. The experimental data (cyan circles) nicely match the results from the NGSPICE (dashed black line) showing the predicted behavior $\Delta f_+ \propto \sqrt{\delta V}$ from TCMT, see Fig. 3c. This sublinear response offers an opportunity to develop an enhanced sensing protocol for detecting small voltage variations, $\delta V$, using the coupling capacitor $C_\kappa(\delta V)$ as a sensing platform. At the same time, the square-root SLR extends the dynamical range (DR) of the sensing measurements up to relatively large values of $\delta V$. The DR is the other important metric that characterizes the efficiency of a sensor, and it is defined as the ratio between the maximum and the minimum $\delta V$ variation that the sensor can measure. Furthermore, the presence of gain elements guarantees the narrowing of the emission peaks and promotes an enhanced resolution. To further quantify the efficiency of our sensing protocol, we have introduced the sensitivity, $\chi \equiv \frac{\partial\left(\frac{\Delta f}{f_0}\right)}{\partial(\delta V)}$. In Fig. 3d we report the measured sensitivity (violet circles) together with the NGSPICE results (black dashed line). We find that, within the experimental resolution, $\chi \sim \frac{1}{\sqrt{\delta V}}$ in the proximity of the NLEPD. Obviously, the frequency smoothening of $\Delta f_+$ around the NLEPD (see Fig. 3b), whose origin is traced back to small unavoidable resonant mismatch between the resonances of the two RLC tanks (see Supplementary Note 7 and Supplementary Fig. 4), will eventually set a saturation value for the sensitivity.

## Noise analysis

The sublinear frequency detuning Eq. (3) guarantees an enhanced transduction function from the voltage variation to the sensitivity $\chi$. However, it does not, address another important characteristic of high-performance sensors that is related to the precision of the measurements. The latter is identified with the smallest measurable variation in the input signal that can be identified by the sensor due to noise at the output signal.

To better understand the effects of noise in the measurement process, we have analyzed the Allan deviation, $\sigma_{\widetilde{\Delta f_+}}(\tau)$, of the normalized frequency detunings, $\widetilde{\Delta f}_+ \equiv \frac{\Delta f_+}{f_0}$, as a function of the sampling time $\tau$[42,43]. The measured Allan deviation is reported in Fig. 4a for representative $\delta V$ values−both well within the NLEPD-enhanced sensitivity regime and away from it. We observe that as $\delta V$ decreases, and the system approaches towards the NLEPD, the noise increases. To better appreciate the effects of noise in the measured voltage variation, we report in Fig. 4b the normalized Allan deviation, $\sigma_\alpha(\tau) = \sigma_{\widetilde{\Delta f_+}}(\tau)/\chi$ [V]. Our measurements show that the noise grows slower than the signal enhancement as we are approaching the NLEPD. Therefore, we conclude that the proposed protocol can provide an enhanced SNR in the proximity of a NLEPD. Based on the behavior of Allan deviation, we can distinguish different regimes depending on the duration of the sampling time $\tau$. Each regime is influenced by a different type of noise source. At the limiting case of long sampling times, the Allan deviation behaves as $\sigma_\alpha^{DRR} = \alpha_{DRR}\tau$. This is typical of a drift rate ramp (DRR) noise associated with the presence of systematic (deterministic) errors. For intermediate sampling times, the Allan deviation reaches a saturation value $\sigma_\alpha^{BI}(\tau) = \alpha_{BI}\tau^0$, which is indicative of the bias instability (BI) noise. The value of $\alpha_{BI}$ sets the smallest possible reading of our sensor. Its origin is traced to the random flickering of electronics or other components of the system. Finally, the short-time behavior of Allan deviation exhibits voltage (variations) random walk (VRW) noise which decreases with the sampling time $\tau$ as $\sigma_\alpha^{VRW} = \alpha_{VRW} \cdot \tau^{-1/2}$. The origin of the VRW is traced to noise sources, such as thermal (Johnson-Nyquist) noise from the circuit elements (resistors and the amplifier) and attached TLs; the readout noise; and other noise sources associated with fluctuations of the voltage applied to the coupling capacitor or fluctuations of the capacitance due to thermal variations. Each of them is described by a separate noise coefficient $\alpha_{cir}$, $\alpha_{TL}$, $\alpha_{det}$, $\alpha_{add}$, respectively, and contributes to the noise equivalent voltage variation given by $\alpha_{VRW} = \sqrt{\alpha_{JN}^2 + \alpha_{det}^2 + \alpha_{add}^2}$, where $\alpha_{JN}^2 = \alpha_{cir}^2 + \alpha_{TL}^2$. Since $\alpha_{det}$ can be actively minimized, the thermal noise, $\alpha_{JN}$, together with $\alpha_{add}$, represent the best obtainable limit for $\alpha_{VRW}$.

In Fig. 4c, we provide a panorama of $\alpha_{VRW}$ for all $\delta V$-values that we have used in our measurements (blue circles). From the measurements, we conclude that $\alpha_{VRW} \approx 0.002\,\delta V^{0.5} [Vs^{\frac{1}{2}}]$, indicating a robustness to the VRW noise, which is attributed to the stable

hyperbolic nature of the $f_+$ supermode. Specifically, the phase-space around these hyperbolic fixed points are structurally stable[41], and the associated basin of attraction is relatively broad—even for voltage variations close to the NLEPD (see Fig. 2c). Therefore, the VRW noise is not able to significantly push the phase-space trajectories out of the basin of attraction. In the same figure, we also show the noise coefficient $\alpha_{JN}$ from NGSPICE (red diamonds), where we have incorporated thermal noise at the resistors, the amplifier, and the TLs described by an ambient temperature of $T = 300$ K. The simulated value of $\alpha_{JN}$ is over an order of magnitude smaller than our experimental measurements for $\alpha_{VRW}$. We conclude that the detection noise $\alpha_{det}$ combined with $\alpha_{add}$ overwhelms the thermal noise $\alpha_{JN}$. Eventually, the absolute bound of $\alpha_{VRW}$ will be determined by the noise due to coupling fluctuations that originate from applied voltage uncertainties and temperature-dependent capacitive effects.

## Discussion

We have demonstrated a SNE sensing of an NLEPD-based voltmeter. The proposed sensing protocol is based on a square-root frequency detuning from the NLEPD induced by a small voltage variation that modifies the coupling between two nonlinear RLC tanks. The NLEPD occurs at the transition between two types of oscillation quenching regimes, i.e., OD and AD domains, in the parameter space, which are consequences of the nonlinear gain/loss channels assigned to each RLC tank. The underlying phase space is structurally stable in the proximity of the (stable) fixed points associated with the degenerate NS, while the corresponding basins of attraction are relatively broad, even for voltage variations that are close to the NLEPD. These characteristics, shield the sensing signal from noise resulting in a two orders of magnitude enhancement of signal-to-noise ratio in the proximity of the NLEPD. Our results establish EPD-sensing from self-oscillating systems as an efficient platform with a dramatically improved SNE factor in the proximity of the NLEPD. Our scheme can guide the design of oscillation quenching-based hypersensitive sensors with enhanced dynamical range that can be utilized in electro-encephalography, electrocardiography and neuroprosthetics. Other applications that can utilize the current design include sensitive manometers, flow sensors, accelerometers, inclinometers for telemetry[44] and implantable microsensors[45]. Many of these applications could also be realized in other frameworks, such as photonics, where circuits with neuromorphic functionalities have already been reported[46].

Let us finally point out that while the specific form of the nonlinearity used in our design leads to an enhanced signal-to-noise ratio in the proximity of the NLEPD, this is not necessarily the case for just any self-oscillating system. For example, in refs. 13,16, the EPD-based Brillouin ring-laser gyroscope was found to enhance the noise in the proximity of the EPD, offsetting the enhancement in sensitivity. In this respect, it will be interesting to extend the current investigation of SNE sensing schemes to other self-oscillating systems whose nonlinear nature allows for the presence of limit cycles and Hopf bifurcations[47]. Sublinear sensing protocols based on nonlinear mechanisms are a promising direction for building hypersensitive SNE sensors. For example, a recent theoretical proposal that advocates for SNE[48] utilizes sublinear intensity variations due to a dynamic hysteresis occurring in nonlinear resonators. Its experimental implementation, however, could be problematic for fast sensing purposes, since it requires up-down sweeping of a control parameter which can be time-consuming. Further exciting future research includes the identification of other physical observables, such as scattering cross-section anomalies (e.g., Wigner cusps)[49,50] and transmission peak degeneracies[15], whose sublinear response for sensing purposes has thus far been utilized in linear settings.

## Methods

### Circuit design and fabrication

The circuit schematic used for this experiment can be seen in Supplementary Fig. 1. It features two RLC resonators which are coupled to each other via voltage-controlled capacitors. The main elements that make up each RLC resonator are a resistor, $R_i$, an inductor, $L_i$, and a pair of in-parallel grounded capacitors, $C_v$ and $C_i$, where $i = 1,2$, denotes the gain and loss RLC units, respectively. The inductors, $L_i = 200\ \mu H$, used in both the gain and loss resonators are API Delevan 807-1537-90HTR. The total capacitance in each resonator is made up by a combination of a tunable capacitor, $C_v$, connected in parallel with a fixed capacitor, $C_i$. The tunable capacitor, $C_v$, is a Murata 81-LXRW19V201-058 with a capacitance range of 100–200 pF. A voltage of 0.5 V was applied to the tunable capacitor in each RLC resonator, and it was controlled by a EG&G Instruments 7265 DSP lock-in amplifier, that was connected via a Bayonet Neill–Concelman (BNC) port to a resistor, $R_v$, model Yageo 603-RC0402FR-074K99L, with a resistance of $R_v = 4.99$ kΩ. This resistor was connected to a grounded fixed capacitor, $C_{v1}$, a Murata 81-GCM32EL8EH106KA7L, with a capacitance of $C_{v1} = 10\ \mu F$. The fixed capacitors in each resonator unit, $C_i$, is a Kemet 80-C0603C911F5G with a capacitance of 910 pF. This gives a total capacitance in each RLC resonator of 200 pF + 910 pF = 1110 pF.

Each RLC resonator has resistive elements that collectively provide gain and loss. The former is provided by an operational amplifier (op-amp) model Analog Devices 584-ADA4862-3YRZ-R7. The power supply for the op-amp was connected by a standard 3 pin connector, with one going to ground, and the other two being connected to $V_+ = 6$ V and $V_- = -6$ V, respectively. To produce gain, the op-amp has a pair of internal resistances, $R_{G1}$ and $R_{G2}$, of 550 Ω each. $R_{G1}$ is connected in between the output of the op-amp and the inverting input of the op-amp. $R_{G2}$ is connected on one end to the inverting input of the op-amp and, grounded on the other end. The mechanically tunable variable resistor in the gain RLC tank, $R_1$, is a Vishay 71-PHPA1206E2001BST1 component that has a resistance of $R_1 = 2000$ Ω. $R_1$ is connected on one end to the operational amplifier's non-inverting input. The other end of $R_1$ is connected to a capacitor, $C_{e1}$, model Kemet 80-C0805C100FDTACTU where $C_{e1} = 10$ pF, capacitively couples each RLC resonator to transmission lines. A mechanically tunable variable resistor, $R_{T1}$, is connected in parallel to $R_1$, and the op-amp, is a Bourns 652-3269W-1-102GLF with $R_{Ti} = 720 \pm 200$ Ω. The set value of $R_{T1} = 720$ Ω, and it is connected to a pair of grounded back-to-back of diodes, $D_i$, Onsemi 512-1N914BWS—which represent the nonlinear elements of this RLC dimer. The fixed resistor in the loss resonator, $R_2$, is a Vishay 71-PCNM2512E2501BST5 component with a resistance of $R_2 = 2500$ Ω. $R_2$ is connected in parallel to $R_{T2} = 750$ Ω. On the other end, $R_{T2}$ is connected to a pair of back-to-back grounded diodes—the same model of diodes as in the gain resonator was used.

The coupling between the two resonators was achieved using two parallel variable capacitors, $C_{vc}$. The component used in $C_v$, a Murata 81-LXRW19V201-058 is the same model as the variable capacitors in the resonator units, $C_{vc}$. As in the RLC tanks, $C_{vc}$ is connected to a grounded $C_{v1}$ which in turn is connected to $R_v$. The same lock-in amplifier is used to control the tuning voltage via a BNC port. One of the two capacitors is held fixed at 200 pF with a constant applied voltage of 0.5 V, whereas the other is tuned in voltage range between 0.4 and 3.5 V.

### Voltage and frequency detuning measurements

The emitted signal from the electronic circuit was collected for different applied voltage variations of the capacitance voltage control ($C_{VC}$) capacitor. These voltage variations where electronically controlled via an EG&G Instruments 7265 DSP lock-in amplifier. The imposed voltage variations were in the range between 0.4 and 3.5 V

associated with $-1.1\,V \le \delta V \le 2\,V$ with resolution of up to 1 mV. At each specific voltage variation, the emitted spectrum was collected using a network analyzer Keysight E5080A. The individual frequency sweeps contain 4001 points in a range of 295–320 kHz. A single measurement was obtained from the collected spectrum with an intermediate frequency bandwidth (IFBW) of 100 Hz giving a sampling time of 40.01 s. The peak frequencies of the spectrum $f_+$ were then identified from the resulted spectrum, which allows the calculation of the frequency detuning $\Delta f_+$.

### Allan deviation measurement

The Allan deviation $\sigma_{\widetilde{\Delta f_+}}(\tau)$ of the frequency associated with the voltage power spectrum peak of the emitted signal is defined as

$$\sigma_{\widetilde{\Delta f_+}}(\tau) = \sqrt{\frac{1}{2(M-1)}\sum_{n=1}^{M-1}\left(\left\langle\widetilde{\Delta f}_+^{(n+1)}\right\rangle - \left\langle\widetilde{\Delta f}_+^{(n)}\right\rangle\right)^2}; \qquad (4)$$

where $\tau$ is the sampling time, $M$ is the total number of frequency measurements and $\langle\widetilde{\Delta f}_+^{(n)}\rangle$ indicates the average rescaled frequency detuning during the sampling time interval $[n\tau,(n+1)\tau]$. For the extraction of Allan deviation, the rescaled emitted peak detunings $\widetilde{\Delta f}_+ \equiv \frac{\Delta f_+}{f_0}$ were sampled with an IFBW of 10 kHz for 101 points in a frequency range of 7 kHz centered around the expected $\Delta f_+$ for the associated $\delta V$. Twenty thousand consecutive spectral measurements were performed over a period of approximately 2700 s for each $\delta V$. This results in a sampling time of 0.1337 s.

### NGSPICE simulations

We use NGSPICE, an open-source software for electronic circuits, to simulate the dynamical behavior of our experimental platform. We consider two $RLC$ tanks coupled by a capacitor using the same characteristics as the experimental platform, where, unless specified otherwise, uses the same parameters of the electronic components described in "Circuit Design and Fabrication" sub-section of "Methods." The op-amp in the gain resonator is represented by a high impedance Norton amplifier by designating its constituent components—a transconductance that quantifies gain, a capacitor and diode clippers. Nonlinearity in the gain and loss resonators is modeled with back-to-back diodes via 1N914 diodes—the same type as used in the experimental platform—using the appropriate parameters that describe its behavior. To compensate for the detuning due to the capacitor in the op-amp, the capacitor in the gain resonator, $C_v + C_1$, was detuned slightly to $0.9955(C_v + C_1)$, to fit the experimental data obtained. In conjunction with that $L_i$ ($i = 1,2$) was set at $0.965L_i$, in the simulations. Using this setup, we evaluate the signal generated by the circuit for a total time of $t = \frac{6000}{f_0} \approx 0.017\,s$ with the time steps of $dt = \frac{1}{(80f_0)} \approx 37\,ns$. In our analysis of the power spectrum, we dropped the first 0.0017 s, which correspond to the short time transient, to consider only the steady-state behavior.

To add noise, we modeled Johnson–Nyquist noise—electronic noise due to thermal fluctuations. This was added to our simulations by adding random voltage sources at every resistor in our circuit (including the TLs). The amount of noise added at each resistor using the TRNOISE function of NGSPICE at these resistors are dictated by the root mean square of the voltage due to Johnson–Nyquist noise. This is given by $v_{rms} = \sqrt{4k_B TRB}$ where $k_B$ is Boltzmann's constant; $R$ is the value of the respective resistor at which the random voltage source was added; $B$ is the bandwidth of the noise for which the natural frequency of the resonator $\approx 338$ kHz was assumed; and $T$ is the temperature for which the ambient value of 300 K was chosen.

## Data availability

The datasets generated during and/or analyzed during the current study are available in the Zenodo repository[51] (https://doi.org/10.5281/zenodo.8250657).

## Code availability

The codes for reproducing TCMT data are deposited at the Zenodo repository[51] (https://doi.org/10.5281/zenodo.8250657).

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

## Acknowledgements

A.S., R.K., and T.K. acknowledge partial support from MPS Simons Collaboration via grant No. 733698 and from NSF-CMMI-1925543. A.S. and T.K. also acknowledges partial support from grant NSF ECCS 2148318, which is supported in part by funds from OUSD R&E, NIST, and industry partners as specified in the Resilient & Intelligent NextG Systems (RINGS) program. L.F.-A. acknowledges support by CONICET Grant No. PIP2021 (11220200100170CO). We acknowledge useful discussions with Professor U. Kuhl on improvements of noise analysis, Professor F. Ellis on circuit design and Mr. W. Tuxbury for assisting with the experimental platform. We also acknowledge useful discussions with Professor G. Aaron and Professor K. Perks on neuronal functionalities.

## Author contributions

A.S. and R.K. designed and fabricated the electronic circuit. A.S. and R.K. characterized the electronic circuit and performed the experimental measurements. A.S. and L.F.-A. developed the theory and carried out simulations and data analysis. T.K. formulated the project. All authors discussed the results. A.S., L.F.-A., and T.K. wrote the manuscript with input from R.K.

## Competing interests

The authors declare no competing interests.
