## [Peer Review File · Nature Communications]

REVIEWER COMMENTS

Reviewer #1 (Remarks to the Author):

The manuscript entitled "Noise Resilient Exceptional-Point Sensing Based on Neuromorphic Functionalities" addresses a highly vexing subject associated with the signal-to-noise ratio (SNR) of a parity time-symmetric sensor in operating in the proximity of an exceptional point (EP) degeneracy under auto-oscillating conditions. In this case, nonlinearities are crucial and dictate the dynamics of the system. The authors concluded in favor of such nonlinear EP (NLEP) schemes as far as SNR figure of merit is concerned.

The experimental setup consists of a pair of properly coupled nonlinear RLC resonators. The sensing platform is the coupling capacitor between the two RLC tanks that is voltage-controlled. They show, both theoretically and experimentally, a detuning of the auto-oscillation frequency which follows a square-root scaling with respect to the coupling perturbation – as expected for systems in the proximity of an exceptional point of order 2. Their theoretical analysis utilizes an rudimentary methods of nonlinear dynamics and bifurcation theory. I also, we find instructive the connections with existing literature on oscillation quenching mechanisms.

Furthermore, they conducted a thorough experimental and computational analysis of the effects of noise on the response function of the sensor by investigating the Allan deviation in proximity to, and away from, the EP. They find that the sensor's sensitivity largely overcomes the impact of noise. The ramification of such conclusions in other auto-oscillating sensing platforms is crucial for determining the viability of EP-auto-oscillation schemes.

Furthermore, they conducted a thorough experimental and computational analysis of the effects of noise on the response function of the sensor by investigating the Allan deviation in proximity to, and away from, the EP. They find that the sensor's sensitivity largely overcomes the impact of noise. The ramification of such conclusions in other auto-oscillating sensing platforms is crucial for determining the viability of EP-auto-oscillation schemes.

(a1) We believe that this study from the technological point of view is not properly emphasized and promoted on the title of this manuscript. The authors should strengthen the importance of the applications as a hypersensitive voltmeter a point as they raise in the abstract of this article. Other applications include a sensitive manometer, accelerometer, inclinometer etc. These applications can be briefly discussed in the conclusion section and pertinent references.

(a3) Inset of figure 2. The NGSPICE and TCMT data are not in perfect agreement with one another. Why? Also, why the NGSPICE calculations describe the experimental data so well and the TCMT does not?

4) Fig 2B. The borders of the basins of attraction at phases $\pm \pi/2$, are they abrupt transitions or there is a smooth transition that it is not captured by the simulations?

5) Fig 3B: Again, there is a difference between the predictions of CMT and the NGSPICE. The latter agrees very well with the experimental data while the former shows deviations. The authors should comment on this.

6) Fig 4: There are no error bars in the curves in Figs 4A, 4B, and the theoretical NGSPICE curve in Fig 4C. The authors should provide error bars associated with their measurements, or at least indicative error bars, as they do in Fig 4 C.

7) The equations S12 in the “polar form” presented in the supplement look familiar and used already in another framework. What is the difference between the model in this manuscript and other models, like the one used in [IEEE Antennas and Propagation Magazine 63, 51 (2021)] where numerically limit cycles were computed and a Hopf bifurcation line is analytically calculated. The authors should explain and clarify if their system is operating in such parameters has a Hopf bifurcation point?.

8) The use of alternative sensing schemes that exploit a square root response (or other sublinear behavior) – although not associated to EP degeneracies-- have been already suggested in literature, e.g. in Phys. Rev. Lett. 129, 013901 (2022) or even in the authors’ own paper Nature 607, 697 (2022). The authors should comment on these issues and discuss the implications.

9) The authors do not provide, the eigenvalues of the Jacobian that characterize the stability of the solutions in the domains I to II, where they focus their study. For completeness of the discussion, this information must be provided.

10) In relation to the previous point, the supplementary figure S3 B is crucial, since it demonstrates the stability of the solutions at the EP. I strongly recommend that such a figure should be moved to the main text.

11) The authors claim that a detuning between the resonant frequencies of the RLC resonators destroys the bistability close to the EP favoring the upper fixed point with higher frequency. The authors have to clarify this point.

12) The authors claim that a detuning between the resonant frequencies of the RLC resonators destroys the bistability close to the EP favoring the upper fixed point with higher frequency. The authors must clarify this point.

Finally, I strongly recommend that this article is published after all corrections were made

Reviewer #2 (Remarks to the Author):

The paper entitled “Noise Resilient Exceptional-Point Sensing based on Neuromorphic functionalities” is timely and it is addressing from theoretical and experimental perspective an important fundamental and technological question related to the efficiency of exceptional point sensors as far as noise effects are concerned.

The authors have developed an electronic circuit platform that operates as a sensitive voltmeter. They bring the system in the proximity of a non-linear exceptional point degeneracy (NLEPD) and measure the sensitivity and signal-to-noise ratio associated with the detuning of the self-oscillation frequency from the NLEPD frequency. They have demonstrated two-orders of magnitude enhancement when the system is in the proximity of the NLEPD as compared with the values that they get when the system parameters are such that it is away from the NLEPD.

The analysis is scientifically sound, and the reported results and conclusions are interesting for a broad audience. The proposed platform can be used for a variety of applications as the authors indicate at the conclusions. I recommend publication to Nature Communications, provided that the authors address my questions below.

1) It seems that the main result of this paper is opposite to the conclusions of Vahala and Langbein reported in [13], [16], [18]. In these cases, the used platforms consist of coupled pair of lasing / self-oscillating modes which emit signal with finite amplitude, implying that in all cases saturable nonlinearities are present and EPDs are in fact NLEPDs. However, the noise behavior reported in this

paper is very different to the one observed in the optical platforms by Vahala and Langbein. Can the authors clarify the aspects that fundamentally differentiate the proposed sensing platform from the one studied by K. Vahala and W. Langbein?

2) Fig 2a suggests that the amplitude has highly nonlinear behavior in the proximity of NLEPD. Can the amplitude variations be expanded in Puiseux series? If so, can the platform be tuned such that those variations would be happening around zero amplitude point?

3) I think the title of Fig 3 is somewhat misleading, since panel A does not show the experimental results. It would be useful to add theory/experiment labels to the panels to improve the readability.

4) Fig 3c suggests that sensitivity is diverging around the NLEPD ($\Delta V=0$), while measured emitted spectrum reported in Fig 3b and NGSPICE simulations clearly show that the singularity is somewhat washed away for tiny, small ΔV . From the plot I see that the signal has finite nonvertical slope at $\Delta V=0$. Could the authors explain this point? Also, it would be useful to mention in the main text if there is any sensitivity saturation observed.

5) What dictates the experimental platform limitations to measure tiny detuning's?

6) Some references to the Allan deviation and its different regimes would be useful, since this concept is more common in engineering rather than in scientific community.

7) How the error bars in fig 2a are calculated? What does the "error analysis" mean?

Reviewer #3 (Remarks to the Author):

In this manuscript, Suntharalingam et al. investigated the utilization of oscillation quenching phenomena for sensing devices. The goal of this study is to overcome the noise-sensitive defects of the sensing techniques based on exceptional point degeneracies, by exploiting oscillation quenching phenomena that are topologically protected in terms of dynamical theory. The authors demonstrated their proposal through experimental demonstrations utilizing RLC circuits.

To the best of the reviewer's knowledge, the most significant impact of this work is the first experimental demonstration of oscillation quenching phenomena (oscillation death and amplitude death) in non-Hermitian systems that exhibit parity-time (PT) symmetry in the linear regime. Employing the observed phenomena for sensing applications is both highly appropriate and timely, as this approach addresses the trade-off issue between sensing sensitivity and operation resilience, which is inherently challenging to overcome with linear systems. Consequently, the reviewer is convinced that the novelty and impact of this study meet the stringent criteria of Nature Communications.

However, certain conceptual descriptions, theoretical terminologies, and connections to previous studies appear ambiguous and confusing. In order to clarify the impact of this work, a thorough revision is crucial. The theoretical and experimental analysis conducted in this work is also technically sound. Consequently, the reviewer recommends a major revision of the manuscript at this stage.

1. In the title, the authors asserted that oscillation quenching phenomena represent neuromorphic functionalities. However, such interpretations may raise some confusion in the current form of the manuscript. It is correct that oscillation quenching phenomena can be found in biological systems including nervous systems, e.g. the interaction between nonlinear ionic channels. However, not all oscillation quenching phenomena are neuromorphic because they are observed in general dynamical systems. Therefore, the claim for neuromorphic behaviors may require a more clarified connection to neuronal behaviors, e.g. the similarity between the nonlinearities in this work and in neural systems. The reviewer therefore suggests the inclusion of such a comparison, or the revision of the manuscript focusing on oscillation quenching phenomena.

2. A similar confusion is also present in the Introduction section. While introducing the concept of oscillation quenching phenomena, the authors described the oscillation death (OD) and amplitude death (AD) in conjunction with the concept of PT symmetry. This may be confusing, as the OD and AD are more general concepts that can be achieved in dynamical systems without PT symmetry. The description of OD and AD using the general definitions found in ref. [30] would be more helpful for general readers.

3. The connections to previous literature should be more clearly established. The first theoretical finding of oscillation quenching (AD and OD) phenomena in PT-symmetric systems with respect to exceptional points was explored in [26]. The applications of AD and OD phenomena to signal processing [27] and memory devices [Adv. Elec. Mater. 8, 2200579 (2022)] were also theoretically studied. The theoretical proposal of achieving AD and OD using electrical circuits was also demonstrated in [Nonlinear Dyn. 100, 1629 (2020)]. Although the experimental verification presented in the authors' work offers significant impact, it is essential to duly acknowledge such prior achievements.

4. In the theoretical analysis, the authors employed various approximations for a concise description: rotating wave approximation, eliminating fast-oscillating terms, weak coupling, and high-quality factor, which are not entirely independent. These approximations appear to be the source of the discrepancy

with the SPICE model. Including additional discussion on this aspect in the main text, along with some quantitative descriptions of the necessary conditions for each circuit element, would be beneficial for reproducibility.

5. Considering the importance of the field of photonics in non-Hermitian phenomena, the reviewer wonders whether the result of this manuscript can be extended into the regime of photonics. Some discussion may be included in the Discussion section.

6. Some typos are observed: "Using is representation" in Supplementary Materials.

RESPONSE TO REVIEWER COMMENTS

Response to Reviewer #1:

1) The manuscript entitled "Noise Resilient Exceptional-Point Sensing Based on Neuromorphic Functionalities" addresses a highly vexing subject associated with the signal-to-noise ratio (SNR) of a parity time-symmetric sensor in operating in the proximity of an exceptional point (EP) degeneracy under auto-oscillating conditions..... I also, find instructive the connections with existing literature on oscillation quenching mechanisms.....They find that the sensor's sensitivity largely overcomes the impact of noise. The ramification of such conclusions in other auto-oscillating sensing platforms is crucial for determining the viability of EP-auto-oscillation schemes.

→We thank the referee for the positive assessment of our work. We share with them their opinion about the potential ramifications on the viability of EP-auto-oscillation schemes

2) We believe that this study from the technological point of view is not properly emphasized and promoted on the title of this manuscript. The authors should strengthen the importance of the applications as a hypersensitive voltmeter a point as they raise in the abstract of this article. Other applications include a sensitive manometer, accelerometer, inclinometer etc. These applications can be briefly discussed in the conclusion section and pertinent references.

→In the revised manuscript we have emphasized the importance of the applications of our platform by changing the title to "Noise Resilient Exceptional-Point Voltmeters enabled by Oscillation Quenching Phenomena". We have also strengthened the discussion on potential applications in the Conclusions section.

3) Inset of figure 2. The NGSPICE and TCMT data are not in perfect agreement with one another. Why? Also, why the NGSPICE calculations describe the experimental data so well and the TCMT does not?

→The TCMT has limitations. It is an approximated method that works very well in the case of weak coupling between the two RLC tanks, and for high-Q factors of each resonator. Moreover, the TCMT uses values of the resonant frequencies that are slightly detuned (e.g. due to parasitic capacitance in the op-amp) from the experimental values. The observed differences between the prediction of the TCMT and our experimental measurements and NGSPICE simulations are associated with these deviations. Nevertheless, the overall picture/agreement is maintained. In the revised version of our Supplementary Information (please see last section), we have addressed the origin of the deviations. We have also highlighted these reasons in the main text (please see second last paragraph in Section "Theoretical Analysis of Oscillation Quenching and Characterization of Nonlinear Supermodes" in page 8).

4) Fig 2B. The borders of the basins of attraction at phases $\varphi = \pm\pi/2$, are they abrupt transitions or there is a smooth transition that it is not captured by the simulations?

→Our TCMT analysis indicates that the borders of the basins of attraction correspond to sharp transitions at initial relative phases $\varphi(0) = \pm\pi/2$. To clarify this issue, we include now in the supplement a new section where we discuss these transitions via a mathematical argument. The main idea behind this argument is that the eigenvector with the largest initial amplitude will eventually dominate the dynamics. Next, we calculate the projection of the initial state against the nonlinear supermodes and we show that the main amplitude switches when the initial relative phase takes values $\varphi(0) = \pm\pi/2$, hence determining the edges of the basin of attraction.

5) Fig 3B: Again, there is a difference between the predictions of CMT and the NGSPICE. The latter agrees very well with the experimental data while the former shows deviations. The authors should comment on this.

→The TCMT uses parameters that corresponds to an ideal circuit leading to sharp curves close to NLEPDs. Instead, the NGSPICE (and of course the experiment) incorporates unavoidable detunings from the ideal NLEPD parameters. We have found that a small detuning of the resonant frequencies of the resonators (due to parasitic capacitance in the op-amp) is responsible for the smoothening of the NS frequencies close to the NLEPD. In the

supplement, we added a new section where we discuss these effects (please see also our previous response). At the same time we also discuss the origin of these deviations in the main text (please see second paragraph before the end of section “Theoretical Analysis of Oscillation Quenching and Characterization of Nonlinear Supermodes”).

6) Fig 4: There are no error bars in the curves in Figs 4A, 4B, and the theoretical NGSPICE curve in Fig 4C. The authors should provide error bars associated with their measurements, or at least indicative error bars, as they do in Fig 4 C.

→ In the revised manuscript we have added error bars in all curves in Fig. 4.

7) The equations S12 in the “polar form” presented in the supplement look familiar and used already in another framework. What is the difference between the model in this manuscript and other models, like the one used in [IEEE Antennas and Propagation Magazine 63, 51 (2021)] where numerically limit cycles were computed and a Hopf bifurcation line is analytically calculated. The authors should explain and clarify if their system is operating in such parameters has a Hopf bifurcation point?

→ The nonlinearity used in the mentioned reference differs from the one used in our work. An important consequence is the existence of limit cycles whereas in our setup we only have fixed points. We thank the referee for pointing out this very interesting work. It will be certainly interesting to see how our conclusions are extended to systems that show limit cycles. We comment this further in the conclusions.

8) The use of alternative sensing schemes that exploit a square root response (or other sublinear behavior) – although not associated to EP degeneracies-- have been already suggested in literature, e.g. in Phys. Rev. Lett. 129, 013901 (2022) or even in the authors’ own paper Nature 607, 697 (2022). The authors should comment on these issues and discuss the implications.

→ Indeed, other nonlinear platforms have been suggested for sublinear sensing, like the ones mentioned by the referee. As far as the PRL is concerned, such a system does not rely on exceptional point degeneracies at all, but on the bi-stabilities that arise in the associated dynamical system. This is a dynamical effect, associated with a hysteresis. In order to exploit the two branches, one needs sweeping up-and-down in some parameter – a process that is time consuming. Furthermore, the suggested paper is a theoretical work and is yet to be demonstrated experimentally. On the other hand, in the Kononchuk et al., Nature 2022, the authors circumvent the problem of noise in the proximity of EPs in linear systems by taking advantage of an associated sublinear response associated with the transmission peak degeneracies (TPD) occurring in the proximity of the standard EPD of the system. The used protocol relies on transmission measurements as opposed to our current proposal which utilizing “lasing action” and relies on nonlinearities to stabilize the system and enhance the signal to noise ratio. We have extended our conclusions to discuss such protocols and contrast them with the current proposal.

9) The authors do not provide, the eigenvalues of the Jacobian that characterize the stability of the solutions in the domains I to II, where they focus their study. For completeness of the discussion, this information must be provided.

→ We now strengthen the discussion of the stability analysis by providing, in the new Fig. 2A, the analysis of the eigenvalues of the Jacobian, evaluated at the nontrivial nonlinear supermodes.

10) In relation to the previous point, the supplementary figure S3 B is crucial, since it demonstrates the stability of the solutions at the EP. I strongly recommend that such a figure should be moved to the main text.

→ In the revised version of the manuscript we modified Fig. 1, to introduce the former Fig. S3B as an inset in Fig. 1B.

11) The authors claim that a detuning between the resonant frequencies of the RLC resonators destroys the bistability close to the EP favoring the upper fixed point with higher frequency. The authors have to clarify this point.

→ In the last section of the revised Supplementary Information, we discuss the effect of a small frequency detuning. We also comment on the impact of the detuning on the stability of the NS. In this respect, a small detuning is able to render the lower branch unstable in the vicinity of the NLEPD and favor the upper branch, which still preserves the square root shape.

12) Finally, I strongly recommend that this article is published after all corrections were made

→ We thank the referee for their positive evaluation of our work.

Response to Reviewer #2

1) The analysis is scientifically sound, and the reported results and conclusions are interesting for a broad audience. The proposed platform can be used for a variety of applications as the authors indicate at the conclusions. I recommend publication to Nature Communications, provided that the authors address my questions below.

→ We thank the referee for their positive evaluation of our work.

2) It seems that the main result of this paper is opposite to the conclusions of Vahala and Langbein reported in [13], [16], [18]. In these cases, the used platforms consist of coupled pair of lasing / self-oscillating modes which emit signal with finite amplitude, implying that in all cases saturable nonlinearities are present and EPDs are in fact NLEPDs. However, the noise behavior reported in this paper is very different to the one observed in the optical platforms by Vahala and Langbein. Can the authors clarify the aspects that fundamentally differentiate the proposed sensing platform from the one studied by K. Vahala and W. Langbein?

→ In the paper by Langbein, the author used a transmission measurement protocol in the proximity of an EPD. This has to be contrasted with our case that we use a self-oscillating setup. On the other hand, it is a fact that in both Vahala's and our scenario nonlinearities are used to obtain stable self-oscillations. However, different types of nonlinearities can lead to very different stability behavior of the system. In our case, both nonlinear elements are lossy, and the linear gain is eventually compensated by the nonlinear loss. This has to be contrasted to Vahala's scenario where he uses nonlinear gain. We comment on these differences in the conclusion section.

3) Fig 2a suggests that the amplitude has highly nonlinear behavior in the proximity of NLEPD. Can the amplitude variations be expanded in Puiseux series? If so, can the platform be tuned such that those variations would be happening around zero amplitude point?

→ Thank you for bringing to our attention this possibility. Indeed, the amplitudes have a highly nonlinear behavior in the vicinity of the NLEPD. In fact, the predictions of the CMT indicate that they follow a power law, i.e. $\log\left(\frac{V_1}{V_2}\right) \sim |\delta V|^{2/3}$. A Puiseux expansion yields $\frac{V_1}{V_2} \sim 1 + |\delta V|^{2/3} + O(|\delta V|^{4/3})$, which also reveals a sublinear response. These features suggest that also the amplitudes might be used for sensing, however that falls beyond the scope of the current paper and deserves a further study. Nevertheless, we have commented about this possibility in the revised manuscript (please see second last paragraph in Section "Theoretical Analysis of Oscillation Quenching and Characterization of Nonlinear Supermodes" in page 8).

4) I think the title of Fig 3 is somewhat misleading, since panel A does not show the experimental results. It would be useful to add theory/experiment labels to the panels to improve the readability.

→ The referee is correct. We have changed the title of the figure and indicated clearly in the caption what panels correspond to theory (TCMT) and to experimental measurements.

5) Fig 3c suggests that sensitivity is diverging around the NLEPD ($\delta V=0$), while measured emitted spectrum reported in Fig 3b and NGSPICE simulations clearly show that the singularity is somewhat washed away for tiny,

small δV . From the plot I see that the signal has finite nonvertical slope at $\delta V=0$. Could the authors explain this point? Also, it would be useful to mention in the main text if there is any sensitivity saturation observed.

→ In the revised version of the main text (see last paragraph of the section “Theoretical Analysis of Oscillation Quenching and Characterization of Nonlinear Supermodes”) we commented on the fact that tiny small detuning between the resonant frequencies of the resonators leads to a smoothing of the frequency detuning around NLEPD (a behavior also captured by the NGSPICE simulations). Further clarifications are given at the new section in the Supplementary Information “Impact of a frequency detuning”. Furthermore, in order to better report the position of the NLEPD which is used in order to extract the square-root response we have now introduce an indicative line (see magenta point line in Fig. 3B) that marks the position of the NLEPD at $\delta V = 0$. There, the slope is almost vertical, with a large saturation value controlled by resonant detuning (see above) which is responsible for the smoothing of the frequency variation Δf_+ vs. δV near the NLEPD. We have also added a clarification of this point at the last paragraph of the section “Sensing protocol” of the main text.

6) What dictates the experimental platform limitations to measure tiny detuning's?

→ In our platform, the main limitation to measure tiny detunings is given by the voltage control capacitor, which is limited by 1mVolt accuracy. We have clarified this point in the main text (please see clarification below Eq. (3)).

7) Some references to the Allan deviation and its different regimes would be useful, since this concept is more common in engineering rather than in scientific community.

→ In the revised manuscript, we included the new references [42] and [43]

8) How the error bars in fig 2a are calculated? What does the “error analysis” mean?

→ We did 5 measurements of the voltages V_1 and V_2 at each resonator and we took the standard deviation. The error bars indicate 1 standard deviation. We have indicated this now in the figure caption.

Response to Reviewer #3:

1) To the best of the reviewer's knowledge, the most significant impact of this work is the first experimental demonstration of oscillation quenching phenomena (oscillation death and amplitude death) in non-Hermitian systems that exhibit parity-time (PT) symmetry in the linear regime. Employing the observed phenomena for sensing applications is both highly appropriate and timely, as this approach addresses the trade-off issue between sensing sensitivity and operation resilience, which is inherently challenging to overcome with linear systems. Consequently, the reviewer is convinced that the novelty and impact of this study meet the stringent criteria of Nature Communications.

→ We thank the referee for their positive evaluation of our work and in the revised manuscript we have further strengthen the importance of oscillation quenching phenomena in non-Hermitian systems.

2) In the title, the authors asserted that oscillation quenching phenomena represent neuromorphic functionalities. However, such interpretations may raise some confusion in the current form of the manuscript. It is correct that oscillation quenching phenomena can be found in biological systems including nervous systems, e.g. the interaction between nonlinear ionic channels. However, not all oscillation quenching phenomena are neuromorphic because they are observed in general dynamical systems. Therefore, the claim for neuromorphic behaviors may require a more clarified connection to neuronal behaviors, e.g. the similarity between the nonlinearities in this work and in neural systems. The reviewer therefore suggests the inclusion of such a comparison, or the revision of the manuscript focusing on oscillation quenching phenomena.

→ We agree with the referee that oscillation quenching is much more generic phenomenon. We have, therefore, follow their suggestion to focus on oscillation quenching (and its observation in non-Hermitian systems) and we have revised our title, abstract and introduction accordingly.

3) A similar confusion is also present in the Introduction section. While introducing the concept of oscillation quenching phenomena, the authors described the oscillation death (OD) and amplitude death (AD) in conjunction with the concept of PT symmetry. This may be confusing, as the OD and AD are more general concepts that can be achieved in dynamical systems without PT symmetry. The description of OD and AD using the general definitions found in ref. [30] would be more helpful for general readers.

→ The referee is correct. We have modified the introduction section accordingly.

4) The connections to previous literature should be more clearly established. The first theoretical finding of oscillation quenching (AD and OD) phenomena in PT-symmetric systems with respect to exceptional points was explored in [26]. The applications of AD and OD phenomena to signal processing [27] and memory devices [Adv. Elec. Mater. 8, 2200579 (2022)] were also theoretically studied. The theoretical proposal of achieving AD and OD using electrical circuits was also demonstrated in [Nonlinear Dyn. 100, 1629 (2020)]. Although the experimental verification presented in the authors' work offers significant impact, it is essential to duly acknowledge such prior achievements.

→ In our revised manuscript we have appropriately cited and referred to these important works in the introductory section.

5) In the theoretical analysis, the authors employed various approximations for a concise description: rotating wave approximation, eliminating fast-oscillating terms, weak coupling, and high-quality factor, which are not entirely independent. These approximations appear to be the source of the discrepancy with the SPICE model. Including additional discussion on this aspect in the main text, along with some quantitative descriptions of the necessary conditions for each circuit element, would be beneficial for reproducibility.

→ We agree with the comment of the referee about the approximations involved in the theoretical modeling of CMT. We have clarified all these approximations in the revised version of the main text (see last paragraph of the section "Sensing Protocol" and also two last paragraphs in section "Theoretical Analysis of Oscillation Quenching and Characterization of Nonlinear Supermodes"). We have also added a new section in Supplementary Information (see last section) which further clarifies some additional sources of discrepancy.

6) Considering the importance of the field of photonics in non-Hermitian phenomena, the reviewer wonders whether the result of this manuscript can be extended into the regime of photonics. Some discussion may be included in the Discussion section.

→ In the revised manuscript, we have pointed to this opportunity at the conclusion section on the possible implementation of such scenarios in the photonic framework.

7) Some typos are observed: "Using is representation" in Supplementary Materials.

→ Thank you. We corrected this and other typos in the Supplement and in the main text.

REVIEWERS' COMMENTS

Reviewer #1 (Remarks to the Author):

The authors have adequately addressed all my comments and provided convincing answers to all my questions. The quality and clarity of the paper have improved after incorporating all these comments. The rewriting of the introduction which now provides emphasis on the oscillation quenching phenomenon (as a more generic physical phenomenon) gives a broader perspective to this work and hopefully will motivate others working in the area of nonlinear dynamics to connect with developments in non-Hermitian wave physics.

Also, the potential technological ramifications of this work have been now emphasized appropriately both in the title and in the conclusions. In their revised manuscript, the authors have also sharpened the differences of their physical system and sensing scheme with other existing ones that are reported in the literature. This clarification/distinction has further emphasized the originality of their contribution and highlighted the crucial role of various forms of nonlinearities in the formation of exceptional point degeneracies and noise effects. I strongly recommend the publication of the paper to Nature Communication in its current form.

Also, I think such a manuscript deserves proper promotion from the point of view of applied nonlinear dynamics and the application space of low-noise oscillators and clocks.

Reviewer #2 (Remarks to the Author):

The authors have addressed all the points I and the other Reviewers have raised in the first round of review process. The clarity of the manuscript have increased. I have no further concern. Therefore, I recommend this work for publication without any reservation.

Reviewer #3 (Remarks to the Author):

In this resubmission, the authors provided detailed and comprehensive responses to the comments raised by the reviewers. The revised manuscript, emphasizing the experimental verification of oscillation quenching phenomena in non-Hermitian systems, now accurately describes the unique contributions of their work. In addition, the terminology and concepts related to oscillation death and amplitude death are correctly applied. The discussion for the approximations and the discrepancy between theory and experiment further enhances the completeness of the manuscript. The reviewer thanks the authors' significant efforts during the revision and is pleased to recommend the acceptance of the manuscript.